# Assessment of a multiplex arbovirus PCR Detection Test in an area endemic for Chikungunya, Zika, and Dengue viruses: An evaluation of kit performance characteristics in line with Clinical Laboratory Improvement Amendments (CLIA) Standards

Leonard Kingwara[1]*, Lazarus Odeny[2], Vera Onwonga[1], Rukia Madada[1], Marygorett Mbeneka[1], Jully Okonji[1], Victoria Mwende[1], Charles Rombo[1], Shalyn Akasa[1], Millicent Ndia[1], Fredrick Ouma[1], Isabella Ayagah[1], Abdi Roba[1], Kamene Kimenye[1]

1 Kenya National Public Health Institute, Kenya, 2 Amref Health Africa, Africa

* leonard.kingwara@gmail.com

## Abstract

### Introduction

Several multiplex and singleplex PCR systems exist for the detection of Dengue(DENV), Zika(ZIKV), and Chikungunya(CHIKV) viruses; however, their performance characteristics in East Africa remain unassessed. In this study, we investigated the TaqMan® Arbovirus Triplex Kit, which allows for the simultaneous identification of CHIKV, DENV, and ZIKV viruses in serum samples and singleplex assays, including the RealStar® Chikungunya RT-PCR Kit 2.0, RealStar® Dengue RT-PCR Kit 3.0, and RealStar® Zika Virus RT-PCR Kit 1.0.

### Methods

We used Clinical Laboratory Improvement validation methods and residual specimens sourced from DENV, CHIKV, and ZIKV outbreaks, from Kenya, Brazil, and the Democratic Republic of Congo(DRC) in March 2024. Quality control was ensured through Quality Control for Medical diagnostics (QCMD) commercial positive panels. Gold standard results were established through consensus from multiple kits, including the TaqMan® Arbovirus Triplex Kit, Illumina viral surveillance panel (Illumina VSP), and Altona singleplex kit. Data were analyzed using R to determine specificity, sensitivity, negative predictive value (NPV), positive predictive value (PPV), and diagnostic odds ratio, with probit analysis employed to evaluate the limit of detection.

**Data availability statement:** The data relevant to this study are available from Dryad at DOI:10.5061/dryad.76hdr7t68 (https://doi.org/10.5061/dryad.76hdr7t68).

**Funding:** Government of Kenya, Ministry of Health KNPHI-017, Pandemic System Strengthening Kenya National Public Health Institute, Dr. Leonard Kingwara.

## Results

The TaqMan® Arbovirus Kit showed sensitivity exceeding 95% for all three virus targets: 97.98% for DENV, and 100% for both CHIKV and ZIKV. Simillarly, the RealStar® RT-PCR Kits exhibited sensitivities: 93.94% for CHIKV, 90.91% for DENV, and 100% for ZIKV. All kits demonstrated specificity above 99%, with a positive predictive value (PPV) over 99% and a negative predictive value (NPV) exceeding 97%. Probit analysis revealed that the TaqMan® Arbovirus Kit also had a lower detection limit compared to the RealStar kit for all tested viruses.

## Conclusion

The TaqMan® Arbovirus Kit provides multiplex capabilities and performance similar to the RealStar® CHIKV, DENV, and ZIKV RT-PCR Kits. Consequently, these kits can be used interchangeably for detecting DENV, CHIKV, and ZIKV viruses in outbreak situations.

## Introduction

To aid in the differential diagnosis of DENV, ZIKV, and CHIKV, it is crucial to employ a triplex kit that can deliver PCR results for all three pathogens, given their overlapping clinical symptoms [1]. This diagnostic strategy not only simplifies the testing process but also significantly enhances clinical management and patient care. Simultaneous identification of these arboviruses can considerably shorten the diagnosis time, which is vital in regions experiencing widespread viral transmission [2]. Earlier research has demonstrated various multiplex real-time polymerase chain reaction (RT-PCR) assays capable of concurrently detecting multiple arboviruses, confirming their effectiveness in clinical environments [3,4]. In Kenya, test kits being used to test DENV, ZIKV and CHIKV range from serological-based to PCR-based. Within these, are several multiplex reverse transcription polymerase chain reaction (rRT-PCR) assays capable of detecting ZIKV, CHIKV, and DENV viruses simultaneously [5]. Most newly developed assays build on the achievements of earlier tools, including a validated triplex RT-qPCR procedure and the newly devised TaqMan™ Arbovirus Triplex Kit [2,6] however, there is still a need to ascertain their performance in a setting where these pathogens are endemic. Kenya being an endemic region for DENV and CHIKV, provide a good setup for evaluating both the assay's sensitivity and specificity, as well as its practical effectiveness in real-time clinical scenarios, which could revolutionize arboviral diagnosis in areas heavily affected by these viruses [7,8].

The use of polymerase chain reaction (PCR) for diagnosing arboviral infections, especially for DENV, ZIKV, and CHIKV, offers considerable advantages over traditional serological methods. Although serological tests can be helpful, they often present issues such as cross-reactivity, inconsistent antibody responses, and latency in antibody detection, leading to false negatives, particularly in the initial stages of infection [9]. Conversely, PCR directly identifies viral RNA, allowing for early and

precise detection of infections during crucial acute phases when timely diagnosis is essential for managing patient care and executing urgent public health measures [10–12]. The advancements in multiplex PCR technology provide the capability for simultaneous detection of multiple viruses, optimizing laboratory workflows and improving diagnostic efficiency. Numerous studies have shown the effectiveness of multiplex assays in simultaneously identifying and quantifying ZIKV, DENV, and CHIKV viruses within a single reaction, thereby minimizing reagent expenses and turnaround times [9,12]. The adoption of such assays has demonstrated enhanced performance characteristics, including greater specificity and sensitivity, compared to traditional serological techniques. These features are critical in regions where these viruses circulate concurrently, as they facilitate accurate diagnoses while also contributing to better epidemiological data and resource distribution during outbreaks [10]. In Kenya, Altona Real star PCR kit which is CE approved have majorly been used in times of outbreak

The TaqMan® Arbovirus kit's performance characteristics for the simultaneous detection of DENV, CHIKV, and ZIKV have been validated in other settings through rigorous testing against established standards. Multiple studies reveal that this assay maintains high sensitivity and specificity, effectively differentiating between the three arboviruses in clinical specimens [13]. The TaqMan® Arbovirus assay's sensitivity values are reported to be ≥ 95%, along with a low limit of detection, making it a trustworthy option for early diagnosis in these endemic regions. Furthermore, the ability to identify viral RNA in a single reaction significantly cuts down processing time and resource use compared to traditional methods, typically requiring individual tests for each virus [11,12]. Similarly, the RealStar® singleplex has also been assessed and found to perform well, demonstrating comparable sensitivity and specificity. The assay's streamlined process allows for the concurrent quantification of DENV, CHIKV, and ZIKV with minimal cross-reactivity [14]. Importantly, the Altona kit's performance has been validated in various geographical locations as opposed to the newly developed TaqMan® Arbovirus kit. [14,15] While both kits yield excellent results, the decision between them may ultimately hinge on local resource availability, specific laboratory capabilities, and the clinical context of their application. Continued assessments of both kits across different settings will be essential to delineate their respective benefits and applications in real-world usage.

Given the rise in mosquito-borne viral diseases in Kenya, it is vital to perform a comprehensive study on the performance characteristics of TaqMan® Arbovirus kit designed for detecting CHIKV, ZIKV, and DENV. Understanding aspects such as sensitivity, specificity, negative predictive value, positive predictive value, and limit of detection within the resource limited setting such as Kenya is crucial, as most laboratories do not have equivalent resources available in settings where previous validations have been done. Prior evaluations, like that of the Triplex real-time RT-PCR assay, have shown significant promise for rapid detection and accurate diagnosis of these arboviruses [4,12,15], however, their main limitation have been their applications in endemic settings in an outbreak set-up. Moreover, challenges in deploying multiplex assays in endemic regions, as illustrated by the adaptation of human diagnostic kits for mosquito sampling, further underscore the importance of thorough performance evaluations to guide public health responses [15].

Conducting performance evaluations in Kenya and similar contexts is crucial for determining the practical relevance of existing diagnostic kits under local epidemiological conditions. Establishing comprehensive performance metrics will not only assist health authorities in selecting the most appropriate diagnostic tools but also inform future testing strategies, including assessing whether testing can reliably be conducted in triplex or single-plex formats [13]. In this study, we aimed to assess the performance characteristics of the research use only (RUO) TaqMan® Arbovirus Triplex Kit for the simultaneous detection of CHIKV, DENV, and ZIKV in serum samples and Singleplex assays, such as the RealStar® CHIKV RT-PCR Kit 2.0, RealStar® DENV RT-PCR Kit 3.0, and RealStar® ZIKV RT-PCR Kit 1.0, all of which are CE certified by Altona Diagnostics for similar sample type. The performance characteristics (Sensitivity, specificity, Negative predictive value, positive predictive value and limit of detection) was conducted using well-characterized panels made up of outbreak samples from Kenya, the DRC, and Brazil, as well as various commercial panels.

## Methods

**Samples:** Archived samples were utilized. These samples came from 330 symptomatic individuals during the outbreak and 40 characterised panel from QCMD (quality control for molecular diagnostics). DENV and CHIKV samples were collected in Kenya from September 2022 to April 2023, while 52 ZIKV specimens were obtained from Brazil specimens dated between March and April 2018, with an additional 5 ZIKV samples from the DRC. Kenyan specimens were retrieved from the biorepository at the National Public Health Laboratory and tested between April 6th and 15th, 2024.

We recharacterized the outbreak samples to obtain the Gold standard test results by testing them with both the TaqMan® Arbovirus Kit and Altona Real star assays. Any discrepancies in results were resolved using the Illumina Viral Surveillance Panel assay. We successfully characterized 89 samples positive for Dengue, 89 positive for Chikungunya, 57 positive for Zika, and 100 samples negative for all three viruses.

**Sample processing:** Samples were processed at the Genomics and Virology laboratories of the National Public Health Institute in Nairobi, Kenya, using serum specimens retrieved from a biorepository. RNA was extracted from 200 µL of each serum sample with the MagMAX™ Pathogen RNA/DNA Kit on the KingFisher™ Flex system, following the manufacturer's guidelines The process included incubation at room temperature after thawing, followed by centrifugation and extraction using King fisher flex as per the MagMax extraction protocol. For PCR detection, the BioRad CFX96™ and QuantStudio™ 5 Flex systems were utilized with RealStar® assays and the TaqMan® Arbovirus Kit, adhering to specific cycling conditions. To reduce inter-operator variability, all laboratory staff underwent standardized training, and automated RNA extraction platforms were employed to ensure consistency and minimize handling errors. The strict adherence to the manufacturer procedures were vital in achieving accurate and reliable results regarding the diagnostic performance of the TaqMan® Arbovirus Triplex Kit. The study staff adhered to established protocols for nucleic acid extraction and real-time PCR analysis within the Kit inserts.

**Laboratory quality assurance:** Quality control was upheld during the study through the use of external positive and negative controls from Thermo Fisher Scientific, which guaranteed the testing process's integrity. We also obtained commercial positive and negative samples for DENV, CHIKV, and ZIKV from QCMD. Furthermore, the laboratories conducting the analysis conforms to ISO/IEC 17025:2022 ISO 15189 standards.

**Statistical evaluation of performance characteristics:** We conducted all statistical analyses using R. We evaluated the sensitivity, specificity, and agreement rates of the assays by comparing test results from outbreak and commercial panel samples for both the TaqMan® Arbovirus kit and the RealStar® singleplex assays against the gold standard results. Additionally, we applied a regression model to examine the relationship between Ct cycle and input concentration for both the TaqMan Arbovirus Kit and RealStar® singleplex assays, and subsequently generated probit curves using GraphPad Prism version 9.1.1.

## Results

We established the Gold standard results by reconciling previous clinical findings and resolving all discrepancies between the TaqMan® Arbovirus Kit and Altona Realstar through metagenomics. Against the Gold standard results, the TaqMan® Arbovirus Kit had a sensitivity rate exceeding 95% for all three targets tested (DENV: 0.9798, ZIKV: 1.000, and CHIKV: 1.000). Similarly, the Altona Realstar Kits demonstrated a sensitivity greater than 90% for these targets (DENV: 0.9091, ZIKV: 1.000, and CHIKV: 0.9394). The above results are shared in Table 1 and Fig 1. Both kits

**Table 1. Showing the sensitivity, specificity, PPV, NPV and diagnostic odds ratio for the Triplex Test and Altona Realstar.**

| Platform | Test | Sensitivity | Specificity | PPV | NPV | Diagnostic Odds Ratio |
|---|---|---|---|---|---|---|
| TaqMan® Arbovirus Kit | DENV | 0.9798 | 0.9926 | 0.9798 | 0.9926 | 132.7626 |
| Altona_Realstar | DENV | 0.9091 | 0.9926 | 0.9783 | 0.9676 | 123.1818 |
| TaqMan® Arbovirus Kit | CHIKV | 1 | 1 | 1 | 1 | Inf |
| Altona_Realstar | CHIKV | 0.9394 | 1 | 1 | 0.9783 | Inf |
| TaqMan® Arbovirus Kit | ZIKV | 1 | 1 | 1 | 1 | Inf |
| Altona_Realstar | ZIKV | 1 | 1 | 1 | 1 | Inf |

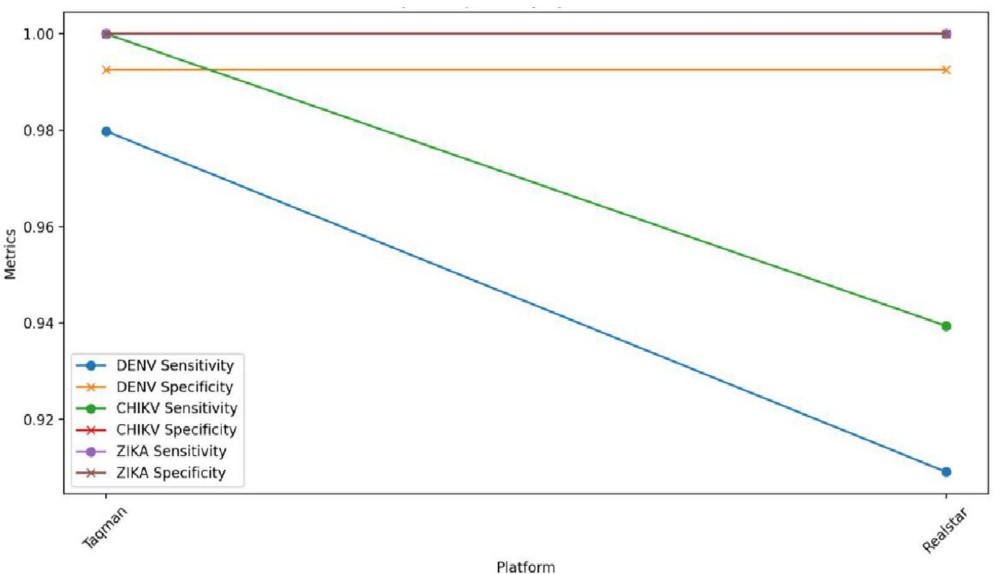

**Fig 1. Sensitivity and specificity by platform and virus.**

exhibited comparable specificity across the three targets. Furthermore, the positive predictive value (PPV), negative predictive value (NPV), and diagnostic odds ratio were similar for all three viruses (DENV, ZIKV, and CHIKV) as shown on Table 1 and Fig 1.

## Comparative sensitivity and spevcificity between the triplex test and Altona Real star

Both kits demonstrated an inverse relationship between concentration and Ct cycle, which aligns with the expected PCR behavior as shown in Fig 2. Probit analysis revealed that the TaqMan® Arbovirus Kit had a lower limit of detection than the Altona Realstar kit for all three targets tested: Zika, Dengue, and Chikungunya as per Fig 3 below.

   Inverse relationship observed across the Altona Realstar and TaqMan Kit tests confirming expected PCR behaviour. TaqMan Kit of detection is comparatively lower on all the three targets(DENV,ZIKV, and CHIKV)

   The probit curve analysis of DENV, CHIKV,and ZIKV, showing the comparative sensitivity of altona Realstar and TaqMan Kit. Across all the three viruses, TaqMan triplex kit was more sensitive

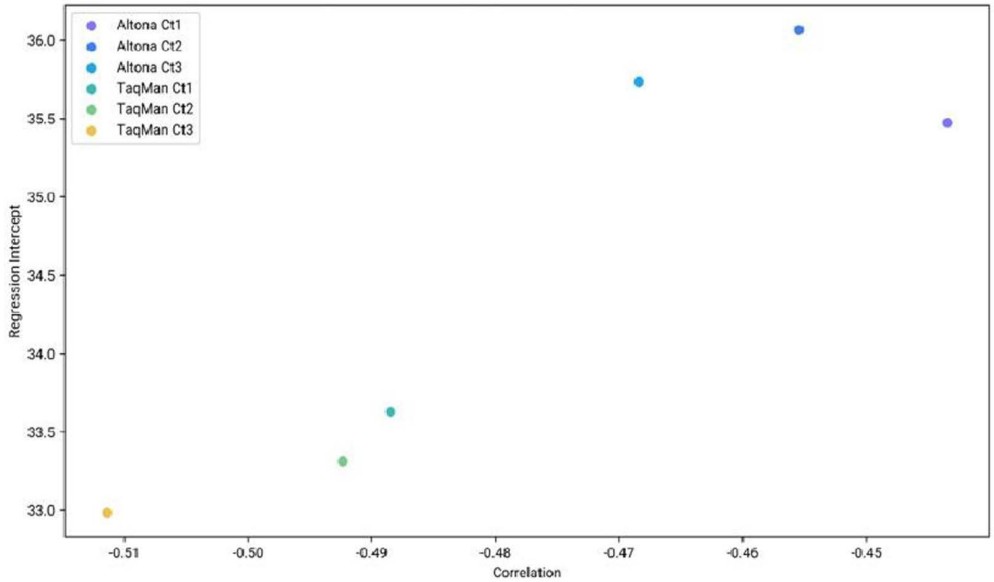

**Fig 2. Releationship between input concentration and ct value.**

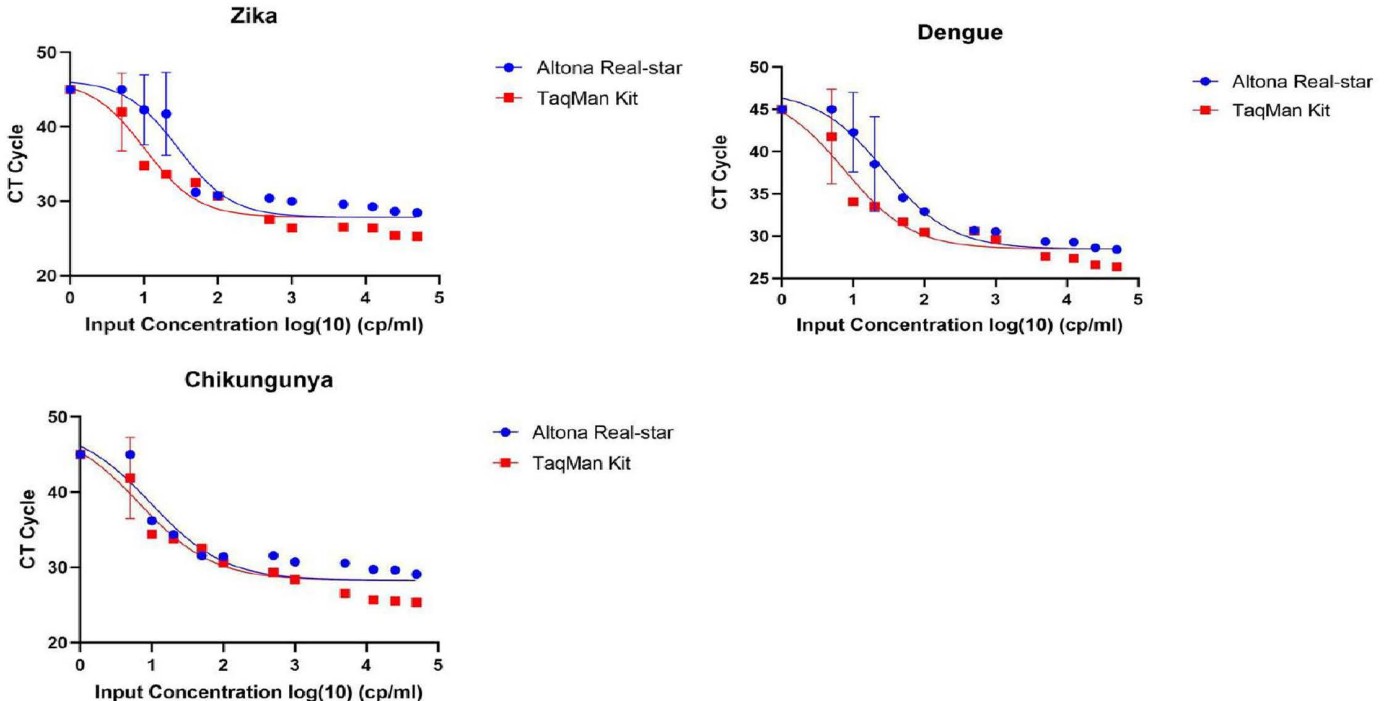

**Fig 3. Probit Curve analysis.**

## Discussion

The TaqMan® Arbovirus Triplex Kit exhibited performance comparable to the CE-certified RealStar® CHIKV RT-PCR Kit 2.0, RealStar® DENV RT-PCR Kit 3.0, and RealStar® ZIKV RT-PCR Kit 1.0 in terms of performance characteristics including sensitivity, specificity, negative predictive value, positive predictive value, and limit of detection. The TaqMan® Arbovirus Triplex Kit demonstrated a lower detection limit in comparison to its sensitivity for DENV and CHIKV, while its results for ZIKV were similar across all kits. Our findings are supported by findings generated by the center for disease control in their validation study [11].Furthermore, both the TaqMan® Arbovirus Triplex Kit and the RealStar® kits showed an inverse relationship between concentration and Ct cycle, which confirmed the expected behavior of PCR and validated the laboratory pipetting procedures as shown in other PCR-validated studies [13].

Although this study has provided a comparison of the performance between the two test kits, it was not specifically designed to generate such results. Therefore, additional studies explicitly designed to obtain comparative data should be conducted. Our research primarily focused on samples collected from Kenya, with ZIKV samples sourced from the DRC and Brazil during an outbreak. In contrast, other comparable studies that have reported similar findings have primarily utilized cultured specimens ([9,12]. While some studies have indeed analyzed outbreak-related clinical specimens, these were conducted outside of East Africa, DRC, and Brazil [10–12], making their conclusions potentially inapplicable to our context due to the varying molecular epidemiological patterns of ZIKV, CHIKV, and DENV across different regions [16–18].

The probit regression analysis has uncovered some unexpected insights. There is a strong correlation between lower cycle threshold values and improved detection sensitivity [19,20]. Infections with CHIKV, ZIKV, and DENV significantly affect positive disease outcomes, underscoring the necessity for early detection and control strategies for these pathogens at the early phase when the viral load is low [20]. As such, the type of test kit used and ZIKV, DENV and CHIKV infection could have a significant influence on disease care in the early phase. Both the TaqMan® Arbovirus Triplex kit and the CE, IVD-approved kit produced similar results, although the TaqMan® kit demonstrated greater sensitivity compared to the Single-Plex RealStar® kit, as shown in our probit curves. Future studies designed with sufficient power could provide more reliable comparative results.

In conclusion, the TaqMan® Arbovirus Triplex kit is comparable to established single-plex RT-PCR test kits for detecting ZIKV, DENV, and CHIKV. Additionally, the sensitivity and specificity for all tested viruses were above 95%. The high level of agreement between the two tests suggests that the new test can be used in areas requiring multi-disease case detection without compromising results. These agreement results further demonstrate that the performance of the TaqMan® Arbovirus Triplex is comparable to that of established Altona RealStar® single-plex assays. One advantage of the TaqMan® Arbovirus assay is its ability to detect the three most common arboviruses in one reaction, reducing variations between experiments as compared to single-plex assays.

## Conclusion

The TaqMan® Arbovirus Triplex Kit can be used as a substitute for the CE-certified RealStar® CHIKV RT-PCR Kit 2.0, RealStar® DENV RT-PCR Kit 3.0, and RealStar® ZIKV RT-PCR Kit 1.0 during an outbreak scenario. A significant benefit of the TaqMan® Arbovirus Triplex Kit is that it provides results for all three viruses simultaneously.

## Limitations of the study

The study was not designed to provide a comparative analysis of the performance between the TaqMan® Arbovirus Triplex Kit and the CE-certified RealStar® CHIKV RT-PCR Kit 2.0, RealStar® DENV RT-PCR Kit 3.0, and RealStar® ZIKV Virus RT-PCR Kit 1.0. Therefore, additional studies should be designed to generate these comparative results if necessary.

## Author contributions

**Conceptualization:** Leonard Kingwara, Rukia Madada, Fredrick Ouma, Isabella Ayagah, Kamene Kimenye.

**Data curation:** Leonard Kingwara, Lazarus Odeny, Jully Okonji, Victoria Mwende, Charles Rombo, Shalyn Akasa, Fredrick Ouma.

**Formal analysis:** Leonard Kingwara, Lazarus Odeny, Vera Onwonga, Rukia Madada, Marygorett Mbeneka, Jully Okonji, Victoria Mwende, Abdi Roba, Kamene Kimenye.

**Funding acquisition:** Leonard Kingwara, Abdi Roba, Fredrick Ouma.

**Investigation:** Leonard Kingwara, Marygorett Mbeneka, Shalyn Akasa, Isabella Ayagah, Kamene Kimenye.

**Methodology:** Leonard Kingwara, Lazarus Odeny, Vera Onwonga, Millicent Ndia, Jully Okonji, Shalyn Akasa.

**Project administration:** Leonard Kingwara, Isabella Ayagah.

**Resources:** Leonard Kingwara, Kamene Kimenye.

**Validation:** Leonard Kingwara, Lazarus Odeny, Rukia Madada, Victoria Mwende.

**Visualization:** Leonard Kingwara, Lazarus Odeny, Millicent Ndia, Jully Okonji, Victoria Mwende, Abdi Roba.

**Writing – original draft:** Leonard Kingwara, Vera Onwonga, Millicent Ndia, Charles Rombo, Shalyn Akasa, Abdi Roba.

**Writing – review & editing:** Leonard Kingwara, Charles Rombo, Shalyn Akasa.

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
