## [Decision Letter · Decision Letter 0]

PONE-D-24-30522Implementation of a Multiplex Arbovirus PCR Detection Assay in an Endemic Setting for Chikungunya, Zika, and Dengue: Performance Characteristics and Field Evaluation Outcomes.PLOS ONE

Dear Dr. Kingwara,

Thank you for submitting your manuscript to PLOS ONE. After careful consideration, we feel that it has merit but does not fully meet PLOS ONE’s publication criteria as it currently stands. Therefore, we invite you to submit a revised version of the manuscript that addresses the points raised during the review process.

We look forward to receiving your revised manuscript.

Kind regards,

Carlos Eduardo Calzavara-Silva

Academic Editor

PLOS ONE

Journal Requirements:

4. We note you have included a table to which you do not refer in the text of your manuscript. Please ensure that you refer to Table 2 in your text; if accepted, production will need this reference to link the reader to the Table.

Reviewers' comments:

Reviewer's Responses to Questions

**Comments to the Author**

1. Is the manuscript technically sound, and do the data support the conclusions?

Reviewer #1: Partly

Reviewer #2: Yes

Reviewer #3: Partly

2. Has the statistical analysis been performed appropriately and rigorously? 

Reviewer #1: No

Reviewer #2: Yes

Reviewer #3: I Don't Know

3. Have the authors made all data underlying the findings in their manuscript fully available?

Reviewer #1: No

Reviewer #2: Yes

Reviewer #3: No

4. Is the manuscript presented in an intelligible fashion and written in standard English?

Reviewer #1: No

Reviewer #2: Yes

Reviewer #3: Yes

5. Review Comments to the Author

Reviewer #1: -Introduction

The manuscript focuses on the molecular detection of CHIKV, DENV, and ZIKV. However, the introduction section exclusively discusses serological tests, with no mention of molecular diagnostics. Given the scope of the study, it is critical to provide a more comprehensive overview of molecular testing approaches, especially as these are more relevant to the core of the article. Additionally, the title implies the development or implementation of a novel multiplex assay, but the study merely evaluates commercially available kits. Clarification is needed in both the introduction and the title to avoid misleading the reader about the nature of the work.

-Methodology

The description of the sample types used in the study is unclear. The "sample processing" section mentions serum, but it is not explicitly stated earlier whether serum or whole blood was used. This should be clarified.

Moreover, ethical approval from the relevant committee should be explicitly mentioned, along with the approval reference number, to ensure compliance with research standards.

The statement, "Methodological enhancements were implemented throughout the study to ensure robustness and reproducibility in the evaluation of the TaqMan® Arbovirus Triplex Kit," is too vague. The specific methodological improvements made should be detailed to allow readers to assess the validity and potential for reproducibility of the results.

RNA extraction procedures should be described in greater detail to enhance transparency, particularly for those readers unfamiliar with the specific methods used in this study.

It is also important to clarify whether the commercial kits used in the study were one-step kits. If so, this should be explicitly mentioned. If not, details about the cDNA synthesis protocol need to be provided.

In the Statistics section, a more detailed explanation of the statistical analyses performed is necessary. This includes besides the tests used, the significance thresholds, and how the data were treated to reach the conclusions presented.

-Results

The current presentation of results is not easy to follow. The use of a graph or visual aid would make the findings more accessible and easier to interpret. This would significantly improve the reader’s ability to understand the data trends and relationships.

Additionally, the results lack critical information regarding the Ct values obtained for each sample across the different kits tested. These values are fundamental to assessing the sensitivity and performance of each assay. Including a detailed table or figure presenting the Ct values for each sample would provide greater clarity and allow for a more thorough comparison of kit performance.

-Discussion

The discussion section lacks a comparative analysis with other multiplex molecular diagnostic methods or kits available on the market. Discussing the results in the context of previous studies or validated multiplex kits would enhance the credibility of the findings and provide a more comprehensive understanding of the study’s contributions to the field.

Reviewer #2: Minor comments:

Methods section: Samples

• The authors need to provide more information about the samples used in the study. Were the samples collected from hospitals, local health centers, or universities, or other locations? The collection were performed during an epidemic event?

• The authors stated that they used 370 samples from symptomatic individuals, which raises questions about the negative group. Could the authors clarify whether this group consisted of patients who were completely negative for all infections, or if they could potentially be positive for other arboviruses or febrile illnesses? Additionally, did the authors have access to the list of symptoms for these patients? If so, it would be beneficial to describe them.

• Were the samples collected outside of Kenya (Brazil and Congo) obtained by the group conducting the study, or were they collected in collaboration with other groups? If the samples were collected through collaborations, it would be helpful to cite the collaborating groups.

Methods section: Sample processing

• For better reader comprehension, I suggest that the last paragraph of this section be added as the final paragraph of the introduction.

• The authors must specify the manufacturers of the PCR kits evaluated in the study in this section. Sometimes it was not clear.

Results: Probit Regression analysis

• Could you explain how the probit regression analysis resulted in the number of viral copies per mL (Lod)? It was not clear for me. Why did the authors not perform a standard curve for that?

Discussion

• Would it be possible to use the tested kit (TaqMan® Arbovirus Triplex Kit) on a large scale in the main testing centers in Kenya? What are the issues regarding cost, availability of equipment, trained personnel, etc.? It would be beneficial to include this information in the discussion.

Reviewer #3: The study addresses an important performance analysis of an assay for the detection of DENV, CHIKV, and ZIKV arboviruses.

Introduction

“Cross-reactivity is particularly rampant in regions endemic to arboviruses with more than one arbovirus in circulation or where vaccination is deployed.” Vaccines for arboviruses are not very common due to their limited availability. Specify in the text which context is being addressed.

The introduction could delve a little deeper into the use of molecular and serological methods for defining arbovirus cases. In the studied region, what is the level of use of serological and molecular tests for case definition?

Methods

“This group comprised 99 individuals who tested positive for DENV, 99 individuals who tested positive for CHIKV, 62 individuals who tested positive for ZIKV, and 110 individuals who tested negative for all three viruses.” It is necessary to specify the methodology or even the kits used in this initial diagnosis to determine the results of these samples. Samples used for method validation must be well-characterized. There is no way to know if the kit/method initially used had adequate performance parameters.

“Specimen were retrieved from the biorepository and tested between April 6th to 15th 2024.” There is no information regarding what this biorepository would be. This data is important, considering that samples from different countries were used, each with its own laws regarding access to these clinical specimens.

“Any discrepancies or discordant results between the TaqMan® Arbovirus Kit and the standardized tests were diligently resolved through Sanger sequencing, creating a strong foundation for the comparative analysis.” What was the method used? There were discrepancies in the test results, and no Sanger sequencing results were shown.

Results

It is not possible to analyze the results of Tables 1, 2, and 3 without the essential information about the samples used in the tests and how the initial results were obtained, as indicated in the Methods section. Do the samples with result discrepancies have high or low Ct values? How can we ensure that the initial results are reliable? In theory, singleplex reactions tend to be more sensitive than multiplex reactions.

Figure 1 – Once again, without the sample information, it is difficult to analyze the obtained results, which show significant differences. The Probit analysis presents copies/mL data that cannot be assessed due to insufficient information.

Discussion

The text does not provide a discussion about the discordant results between the assays. There is no discussion on whether the samples had low, medium, or high genomic loads, as well as the differences observed between singleplex and triplex reactions.

6. PLOS authors have the option to publish the peer review history of their article (what does this mean? ). If published, this will include your full peer review and any attached files.

**Do you want your identity to be public for this peer review?** For information about this choice, including consent withdrawal, please see our Privacy Policy .

Reviewer #1: **Yes: ** Naiara Clemente Tavares

Reviewer #2: No

Reviewer #3: No

---

## [Author Response · Author response to Decision Letter 1]

7 Jan 2025

Sincerest thanks for your response and reviewers’ comments on our manuscript. We sincerely apologize for the great time it has taken us to respond to these comments and hope that a revised version of the manuscript will still be considered by PLOS one. We have modified the paper in response to the extensive and insightful reviewer comments. In the abstract section we have added additional results to give the manuscript proper angulation and clarity to fully address the reviewer’s comments. In addition, we have rewritten sections of the manuscript and revised the title. We will thus not respond point by point as nearly the entire manuscript has been overhauled.

---

## [Decision Letter · Decision Letter 1]

Performance Evaluation of a Multiplex PCR Assay for Chikungunya, Zika, and Dengue Detection: CLIA Standards Assessment in a Partially Endemic Region.

PONE-D-24-30522R1

Dear Dr. Kingwara,

We’re pleased to inform you that your manuscript has been judged scientifically suitable for publication and will be formally accepted for publication once it meets all outstanding technical requirements.

Kind regards,

Carlos Eduardo Calzavara-Silva

Academic Editor

PLOS ONE

Additional Editor Comments (optional):

Dear authors, after the second round of revisions, only minor issues were raised by one of the invited reviewers. I decided to accept the manuscript as long as the you addresses the minor changes (as bellow) to avoid another round of revisions.

Please, see the following minor issues:

1- At some points in the text, the authors write the names of the viruses sometimes with uppercase letters and sometimes with lowercase letters. I suggest that the names be written once, followed by their respective acronyms, and that only the acronyms be used throughout the text.

2- In the conclusion section of the abstract, the word "vírus" should be in the plural form.

3- Improve the resolution of Figures 1 and 2 to enhance the interpretation of the graphs.

4- The captions of all figures need to be more detailed and should also follow a consistent format in terms of font size and color.

5- I suggest combining figures 3, 4, and 5 into a single figure with labels A, B, and C for each of the viruses, and then providing an explanatory subtitle.

6- The acronym used for the Democratic Republic of the Congo should be described in the abstract, not in the discussion.

Please note that the approval of your manuscript is contingent upon the resolution of the above issues.

Reviewers' comments:

Reviewer's Responses to Questions

**Comments to the Author**

1. If the authors have adequately addressed your comments raised in a previous round of review and you feel that this manuscript is now acceptable for publication, you may indicate that here to bypass the “Comments to the Author” section, enter your conflict of interest statement in the “Confidential to Editor” section, and submit your "Accept" recommendation.

Reviewer #1: All comments have been addressed

Reviewer #2: check 1 to 6 issues mentioned above

2. Is the manuscript technically sound, and do the data support the conclusions?

Reviewer #1: Yes

Reviewer #2: Yes

3. Has the statistical analysis been performed appropriately and rigorously? 

Reviewer #1: Yes

Reviewer #2: Yes

4. Have the authors made all data underlying the findings in their manuscript fully available?

Reviewer #1: Yes

Reviewer #2: Yes

5. Is the manuscript presented in an intelligible fashion and written in standard English?

Reviewer #1: Yes

Reviewer #2: No

6. Review Comments to the Author

Reviewer #1: After reviewing the revised manuscript, I noticed that the authors made significant modifications and addressed the suggestions provided. The title, “Assessment of a Multiplex Arbovirus PCR Detection Test in an Area Endemic for Chikungunya, Zika, and Dengue Viruses: An Evaluation of Kit Performance Characteristics in Line with Clinical Laboratory Improvement Amendments (CLIA) Standards,” is much improved compared to the initial version. It now better reflects the scope of the study, as does the revised introduction. My only remaining suggestion is to add captions to the figures to facilitate the interpretation of the graphs.

Reviewer #2: The work entitled “Assessment of a Multiplex Arbovirus PCR Detection Test in an area endemic for Chikungunya, Zika, and Dengue viruses: An evaluation of kit performance characteristics in line with Clinical Laboratory Improvement Amendments (CLIA) Standards” presents interesting results regarding the use of a commercial triplex kit for the simultaneous detection of Dengue, Zika, and Chikungunya viruses in human serum samples collected in Kenya, The Democratic Republic of Congo and Brazil. The performance of the proposed kit is similar to that of kits that detect these viruses individually, making it suitable for use during epidemic periods both in Kenya and other African countries. I suggest that the work be accepted for publication in the Plos One journal after the authors address some minor comments cited above:

1- At some points in the text, the authors write the names of the viruses sometimes with uppercase letters and sometimes with lowercase letters. I suggest that the names be written once, followed by their respective acronyms, and that only the acronyms be used throughout the text.

2- In the conclusion section of the abstract, the word "vírus" should be in the plural form.

3- Improve the resolution of Figures 1 and 2 to enhance the interpretation of the graphs.

4- The captions of all figures need to be more detailed and should also follow a consistent format in terms of font size and color.

5- I suggest combining figures 3, 4, and 5 into a single figure with labels A, B, and C for each of the viruses, and then providing an explanatory subtitle.

6- The acronym used for the Democratic Republic of the Congo should be described in the abstract, not in the discussion.

7. PLOS authors have the option to publish the peer review history of their article (what does this mean? ). If published, this will include your full peer review and any attached files.

**Do you want your identity to be public for this peer review?** For information about this choice, including consent withdrawal, please see our Privacy Policy .

Reviewer #1: **Yes: ** Naiara Clemente Tavares

Reviewer #2: No

---

## [Editor Report · Acceptance letter]

PONE-D-24-30522R1

PLOS ONE

Dear Dr. Kingwara,

I'm pleased to inform you that your manuscript has been deemed suitable for publication in PLOS ONE. Congratulations! Your manuscript is now being handed over to our production team.

Kind regards,

on behalf of

Dr. Carlos Eduardo Calzavara-Silva

Academic Editor

PLOS ONE